# Using an Ultrasound Tissue Phantom Model for Hybrid Training of Deep Learning Models for Shrapnel Detection

**DOI:** 10.3390/jimaging8100270

**Published:** 2022-10-02

**Authors:** Sofia I. Hernandez-Torres, Emily N. Boice, Eric J. Snider

**Affiliations:** U.S. Army Institute of Surgical Research, Joint Base San Antonio Fort Sam Houston, San Antonio, TX 78234, USA

**Keywords:** tissue phantom, ultrasound imaging, shrapnel, detection, deep learning, artificial intelligence, classifier, algorithm, foreign body

## Abstract

Tissue phantoms are important for medical research to reduce the use of animal or human tissue when testing or troubleshooting new devices or technology. Development of machine-learning detection tools that rely on large ultrasound imaging data sets can potentially be streamlined with high quality phantoms that closely mimic important features of biological tissue. Here, we demonstrate how an ultrasound-compliant tissue phantom comprised of multiple layers of gelatin to mimic bone, fat, and muscle tissue types can be used for machine-learning training. This tissue phantom has a heterogeneous composition to introduce tissue level complexity and subject variability in the tissue phantom. Various shrapnel types were inserted into the phantom for ultrasound imaging to supplement swine shrapnel image sets captured for applications such as deep learning algorithms. With a previously developed shrapnel detection algorithm, blind swine test image accuracy reached more than 95% accuracy when training was comprised of 75% tissue phantom images, with the rest being swine images. For comparison, a conventional MobileNetv2 deep learning model was trained with the same training image set and achieved over 90% accuracy in swine predictions. Overall, the tissue phantom demonstrated high performance for developing deep learning models for ultrasound image classification.

## 1. Introduction

Ultrasound (US) imaging is critical in a wide range of medical applications from obstetric procedures to time-sensitive emergency medicine decision support [1]. This emergency medicine use case extends beyond the civilian sector to military medicine where important triage decisions for limited medical evacuation resources could rely on US-based casualty triage [2]. This is especially true in austere medical scenarios or combat casualty care situations where access to higher fidelity imaging modalities may not be possible for up to 72 h post-injury [3,4]. US in these emergency use cases could be used to detect foreign bodies or the aid in the diagnosis of pneumothorax, hemoperitoneum, and other injury states within the extended Focused Assessment with Sonography for Trauma (eFAST) exam [5]. However, access to trained radiology expertise to interpret ultrasound images is likely not feasible in these military or emergency situations. As a result, automation of US image interpretation or classification through development of deep learning models can potentially lower the expertise threshold for making critical triage decisions.

A number of efforts have been focused on deep learning model development for medical image interpretation. These include the detection of SARS-CoV-2 using X-ray images [6,7,8], neurological disease progression in MRI images [9,10,11], and the identification of tumors using CT scans [12,13,14], among others. Advances in foreign body detection have been limited to X-ray imaging techniques identifying pathologies like tuberculosis, however high contrast items in chest cavities like coins, medical devices, jewelry can be confounding [15]. High contrast foreign objects can affect the diagnostic quality of an image [16] and remain an area of concern for training medical diagnosis algorithms. Focusing on ultrasound imaging, review articles have been written on various deep learning model architectures for a range of medical applications [17]. For example, COVID-19 and pneumonia detection have been achieved at close to 90% accuracy from US brightness mode (B-mode) images using an InceptionV3 network [18].

With most neural network architectures and training problems, deep learning algorithms for medical diagnoses require thousands of positive and negative images to properly develop, train, and test algorithms [19]. Collecting these datasets can be inefficient, costly, or impossible if only relying on animal or human image sets. In these scenarios, algorithm development using tissue phantoms may bridge the gap and supplement the need for large quantities of animal or human images sets if the tissue phantom is able to accurately mimic the desired tissue’s echogenicity. The development of various phantom tissues with high contrast backgrounds continues to be a need for the advancement of automated diagnostic methodologies. Review articles have been written on how biomaterials can be used for ultrasound tissue mimicking phantoms [20,21,22]. To support medical research studies, tissue phantoms are cost-effective options for initial studies and training exercises where available resources are scarce [23,24,25,26,27,28]. This is also true for animal studies, where high fidelity tissue phantoms can help reduce the animal testing burden [29,30]. Tissue phantoms have been used for applications such as instrument calibration, manual needle steering, and robotic surgery training, but one current area of interest is medical imaging. Developing a tissue phantom relevant for medical imaging requires adequate complexity to mimic tissue, incorporation of subject variability for creating more robust data sets, and tuning the phantom to the specific medical application.

We have recently shown development of a shrapnel detection convolutional neural network trained from images acquired from a custom-developed tissue phantom [31]. Here, we show how this custom phantom has sufficient heterogeneity and subject variability compared to swine tissue that it can be utilized in a hybridized training format to reduce the animal images required. Further, we highlight the effectiveness of a previously developed ShrapML neural network for this application, in comparison to MobileNetV2 [31]. This results in a shrapnel detection algorithm suitable for US image classification of shrapnel in swine tissue.

## 2. Materials and Methods

### 2.1. Fabrication of a Tissue Phantom Mold 

The thigh mold design was based on circumferential measurements of human adult male thighs and resulted in an average outer diameter of 120 mm. The mold was subdivided into three components: bone, muscle, and fat (Figure 1). The bone had a diameter of 26 mm to mimic an average femoral bone [32,33]. The muscle layer was 94 mm in diameter and the fat layer was the remaining diameter resulting in the fat layer equaling ~ 20% of the total tissue diameter. 3D models were developed using computer aided design software (Autodesk Inventor, San Rafael, CA, USA), converted to STL files, and fabricated using a fused deposition modeling 3D printer (Raise3D, Irvine, CA, USA) with Polylactic Acid (PLA, Raise3D, Irvine, CA, USA) filament. 3D printed parts were finished with a brush on resin coating (XTC-3D, Smooth-On, Macungie, PA, USA) to reduce fluid leaking into the walls of the mold. A muscle layer mold was fabricated with a recess for a nylon bone of proper diameter to slot in the center. The mold was two parts that snap together and were secured by vise-grip and tape to minimize leakage as the tissue phantom solidified. The muscle mold was also fitted with a lid (Figure 1) that was clamped and taped shut after pouring the tissue phantom, as solid components in the phantom fell out of solution if not inverted regularly throughout the solidification process (See next section). An outer fat layer mold was also fabricated with an inner diameter of 120 mm with the same bone recess in the middle. The tissue phantom was created sequentially so that after the muscle layer solidified, it could be placed into the larger fat layer mold to add the outer gelatin layer. A thin plastic film (McMaster Carr, Elmhurst, IL, USA) was cut to the height of each respective mold and placed against the outer wall during solidification to minimize the tissue phantom adhesion to the PLA plastic.

### 2.2. Tissue Phantom Construction Using Gelatin

The phantom made to simulate a human thigh was an adaptation of commonly used gelatin tissue phantoms [26,31], which use gelatin as their bulk component due to the similarity in ultrasound properties to fat and muscle tissue [28]. For our purposes, we used 10% (*w*/*v*) gelatin (Thermo-Fisher, Waltham, MA, USA) dissolved in a 2:1 solution of water and evaporated milk (*v*/*v*) (Costco, Seattle, WA, USA) with different flour (unbleached wheat flour, H-E-B, San Antonio, TX, USA) concentrations per layer. Evaporated milk’s colloidal composition increased echogenic reflection of the tissue phantom compared to using water alone [33]. Evaporated milk has been previously used in tissue-mimicking phantoms to alter the attenuation coefficients to better mimic soft-tissues [34]. Flour has been previously used as an echogenic scattering agent when developing tissue-mimicking phantoms [31,35]. To obtain desired consistency, the water and milk solution was heated to approximately 45 °C, to aid gelatin solubility but not burn the milk. Half of the gelatin mixture was then transferred to another container for adding a different flour concentration. A higher concentration of flour was utilized for the inner layer (muscle), 0.25% *w*/*v*, while a lower concentration was used for the outer layer (fat), 0.10% *w*/*v*. After mixing the flour into solution, the solutions were kept at 45 °C in an oven to maintain the gelatin as a liquid.

Similar to gelatin-based tissue phantoms, agarose-based materials have also been shown to mimic the attenuation coefficients of soft tissue [26,36]. To better simulate the heterogeneous muscle tissue, we incorporated agarose (Fisher Scientific, Fair Lawn, NJ, USA) fragments into the inner layer muscle bulk. A 2% agarose solution (*w*/*v*) in water was created by bringing water to a boil until the agarose dissolved and the solution was held at approximately 65 °C until ready for solidification. Brighter and darker fragments were included in the muscle layer by splitting the 2% agarose solution in half, and adding 1.0% and 0.1% *w*/*v* flour, respectively. These solutions were solidified at 4 °C in 50 mL centrifuge tubes (Fisher Scientific, Fair Lawn, NJ, USA) followed by manual crumbling with a spatula to produce irregularly sized pieces (Figure 2A). Agarose components were then added into the muscle layer mold and then the inner layer gelatin mixture was added (Figure 2B,C). A lid on the inner muscle layer mold allowed inversion every minute for 15 min during solidification to keep the agarose components in suspension in the muscle layer (Figure 2D). After solidification of the muscle layer, the inner tissue phantom with bone was removed from the fabricated mold, placed within the outer layer mold, and the fat-mimicking outer layer was added around the muscle layer (Figure 2E). Solution was held at 4 °C until solidification, approximately one hour. After this time, the completed tissue phantom was removed from the outer mold and used for ultrasound imaging applications within the 48 h of fabrication. The tissue phantom was stored at 4 °C in an airtight container when not in use.

### 2.3. Ultrasound Shrapnel Imaging

Using different iterations of the tissue phantoms, images were collected using a Sonosite Edge (Fujifilm, Bothell, WA, USA) ultrasound instrument and HF50 probe (Fujifilm, Bothell, WA, USA). For simplicity, all ultrasound imaging was performed underwater to avoid the need for ultrasound gel. Baseline images were captured at varying depths and with longitudinal and transverse probe placement in 10 s video clips. Various shrapnel fragments were inserted into the tissue phantom underwater, to avoid air introduction, using surgical forceps. Images were collected for each of 8 shrapnel types (asphalt, ceramic, glass, metal, plastic, rock, rubber, and wood) in each tissue phantom iteration with shrapnel depth varying from 2 cm to 5.5 cm, where it was almost impacting the bone. The entire circumference of the tissue phantom was utilized for shrapnel insertion and imaging.

Similarly, baseline and shrapnel images were collected using recently euthanized animal tissue, e.g., porcine thighs from an unrelated Institutional Animal Care and Use Committee tissue sharing agreement. Similarly, to the tissue phantom, baseline images were obtained at varying depths, angles, and with bone sometimes in view. Subsequently, two lateral 3 cm incisions were made on the bicep and the various types of shrapnel were inserted to depths of 2 to 4.5 cm. An average of 30 images were taken for every shrapnel type, with shrapnel placed at different depths within the tissue and varying angle of imaging. The full collection of ultrasound images (14 iterations of tissue phantoms and swine biceps) was compiled for further image preprocessing. For quantifying image properties, ultrasound images were cropped based on average histogram intensities in the row and column direction to identify the edge of each image using MATLAB R2021b (Mathworks, Natick, MA, USA). Mean and standard deviation values across the cropped images were calculated on grey scale images using MATLAB.

### 2.4. Training Image Classification Algorithms for Shrapnel Detection

Briefly, ultrasound images were imported after cropping to a 512 × 512 size using ImageJ/FIJI [34,35]. Next, 20% of images were held out as a testing image set, and the remaining images were split 80:20 between training and validation. Image augmentation in the form of zoom, rotate, flip, contrast adjustments were performed randomly at magnitudes up to 10%. The classifier algorithm architecture, previously described and termed ShrapML, is shown in Figure 3 and summarized in Table 1. Briefly, it is comprised of a series of 2D convolution layers connected to a rectified linear unit activator with each followed by a max pooling layer. This is repeated 5 times, followed by a dropout regularization layer set at 55% to help reduce model overfitting and then a flattening layer to reduce the image to 1D. A fully connected neural network layer with a sigmoid activator is the final step in the network and results in a binary image classification output: baseline (no shrapnel) or shrapnel. Training was conducted over 100 epochs with a 32-image batch size using an RMSprop optimizer to minimize loss (Table 1). The deep learning image classification algorithm was previously designed using TensorFlow/Keras (version 2.6.0) and Jupyter Notebook in Python (version 3.8) [31]. Training was performed on an HP workstation (Hewlett-Packard, Palo Alto, CA, USA), running Windows 10 Pro (Microsoft, Redmond, WA, USA) with an Intel Xeon W-2123 (3.6 GHz, 4 core, Santa Clara, CA, USA) processor with 64 GB RAM.

The ShrapML classification algorithm was used to demonstrate the functionality of the tissue phantom, specifically how it can be used to reduce or supplement the number of human or animal images required to train an algorithm. We constructed hybrid training data sets of swine images supplemented with different ratios of phantom images, keeping the total image number approximately the same (Table 2). Ratios tested were 1:0, 1:1, 1:3, 1:9, and 0:1 of swine to phantom images. Each ratio was used as the training data set for the algorithm. Training was performed three times with different random validation splits from the total number of images. After training, swine test image sets were used for evaluating trained ShrapML model predictions that were not part of the training data sets (111 unaugmented baseline images, 116 unaugmented shrapnel images). These results were used to calculate performance metrics including accuracy, precision, recall, specificity, and F1 score. The true positive rate, or the rate the classifier model correctly predicted a positive category, was tabulated, along with the false positive rate to generate Receiver Operating Characteristic (ROC) Curves. These ROC curves graphically illustrate how well the classifier model predicts a binary outcome with the dashed diagonal line representing random chance while a perfect classifier would maximize the area under the ROC (AUC) with a true positive rate (or sensitivity) of 1 for all threshold values. Confusion matrices add additional variables, such as the true negative and false negative predictions to visualize if the classifier is confusing classes, i.e., commonly mislabeling negatives as positives. ROC curves, area under the ROC curve (AUC), and confusion matrices were generated using GraphPad Prism 9 (San Diego, CA, USA).

We had previously compared a version of the shrapnel image classification algorithm to more than 10 conventional deep learning models, and identified MobileNetv2 as most comparable in accuracy and computational time for this application [36]. As a result, MobileNetv2 was trained using the same five ratios of swine to phantom hybrid image training sets to compare overall performance. For training with MobileNetV2, however, MATLAB R2021b (Mathworks, Natick, MA, USA) was used to train and test, using the same training options and similar testing setup as shown for ShrapML.

## 3. Results

### 3.1. Overview of the Tissue Phantom for Shrapnel Image Acquisition

The thigh tissue phantom was constructed with gelatin as it was a cost-effective material that is nearly anechoic with its high-water content (Figure 4A). However, higher contrast components can be added to modify the bulk properties of the phantom. For mimicking thigh tissue, the tissue phantom was constructed in two layers around a central bone with different concentrations of flour to show varying ultrasonic scatter at different layers in the phantom. This was suitable for modifying bulk properties of the tissue phantom, but it cannot mimic the heterogeneous complexities observed in physiological thigh tissue (Figure 4C). For that, solidified agarose with two different flour concentrations were suspended in the muscle layer which significantly increased the complexity and created subject variability in the phantom design. The final version of the phantom (Figure 4B) after optimization of these different features had proportional complexity to the swine tissue images (Figure 4C). The ultrasound image heterogeneity was quantified by measuring standard deviation of the pixel intensity across each image (*n* = 10 images each), with swine images having a pixel standard deviation of 30.6 and the final phantom version having a standard deviation of 30.1, a 2% error. Mean pixel intensities were 47.1 for the swine images vs. 41.5 for the final phantom version, a 12% error. While the biological organization of muscle fibers as seen in physiological tissue is not observed in the phantom, the overall complexity of the phantom created a more challenging ultrasound environment compared to traditional tissue phantoms, which is essential in algorithm training scenarios.

### 3.2. Application for Automated Shrapnel Detection

To highlight a deep-learning use case for the tissue phantom developed in this study, the phantom was used to train ShrapML, a deep-learning algorithm for detecting shrapnel in ultrasound images [31]. Various shrapnel shapes, material types, and sizes were embedded in the tissue phantom at different depths and proximities to the bone (Figure 5A,C). With certain shrapnel types and locations in the phantom, the shrapnel was easy to identify via ultrasound. However, the non-uniform complexity of the tissue phantom can make shrapnel identification challenging especially with smaller objects embedded deeper in tissue. This same trend was observed with imaging and shrapnel detection in the porcine thigh subjects embedded with the same shrapnel types (Figure 5B,D). To highlight the location of the shrapnel in each, manually generated segmentation mask overlays are provided (right panels of Figure 5A–D).

### 3.3. Phantom and Swine Training Datasets for ShrapML

With the phantom baseline and shrapnel-containing images acquired, we evaluated if the phantom can reduce the number of animal or human images required to properly train a deep learning image classification algorithm. ShrapML [31] was trained with five swine to phantom image ratios: 1:0, 1:1, 1:3, 1:9, 0:1. This created different use cases where only a fraction of the required images were collected from an animal while the rest of the data set was created from tissue phantom images. Overall accuracy to detect shrapnel in test swine images was as high as 99% in the ShrapML model fully trained on swine image sets (1:0) and at only 61% when trained with only tissue phantom images (0:1), indicating the phantom alone cannot be used to develop a classifier model for animal tissue (Figure 6). When a 1:1 ratio was used, accuracy reached 95% and the AUC was also high (Figure 6A). Supplementing training image sets at a 1:3 ratio of swine images to phantom images, kept accuracy equally high at 96% and ROC analysis had a similar AUC (Figure 6B). Slight increases in false positive and negative rates were detected but performance remained strong, similar to 100% swine image training. With even further supplementation of the swine training image sets with tissue phantom images, the 1:9 ratio model began to drop in accuracy and AUC with average results being 87% and 0.97, respectively (Table 3). However, performance was still much higher compared to the 100% tissue phantom trained scenario. Overall, while accuracy was slightly lower than the 100% swine trained ShrapML model, greater than 95% accuracy with a quarter of the required swine images could represent a much more cost-effective means of developing models such as this.

### 3.4. Ratio Comparison ShrapML vs. MobileNetv2

Results were then compared to MobileNetv2 as its performance was best for this shrapnel image classification as previously determined [36]. Using the same hybrid training ratios, MobileNetv2 performed similarly in 100% swine images, with an accuracy of 0.982 (Table 4). The 100% phantom trained version performed worse when evaluated with swine test images when compared to ShrapML, with an accuracy of 51% and a poor AUC of 0.499 (Figure 7B). At the 1:1 ratio, accuracy remained high at 99%, outperforming ShrapML (95% accuracy), but that trend was reversed for the 1:3 ratio where MobileNetv2′s accuracy dropped to 91% compared to 96% for ShrapML. This downward trend persisted for the 1:9 hybrid training ratio, with ShrapML (87%) outpacing MobileNetv2 (69%) shrapnel prediction accuracy for swine test images.

## 4. Discussion

US imaging can have a critical role in emergency and military medicine if the skill threshold for image acquisition and interpretation can be lowered. US image interpretation has the potential to be automated with properly developed and trained deep learning models. Unfortunately, each deep learning application will require large datasets of condition-specific positive and negative images, which may be impossible to obtain for rare injuries or diseases. To that end, tissue phantoms will be critical for supplementing training sets to lower the need for hard to obtain human or swine image sets.

The tissue phantom detailed here for shrapnel applications is a multi-layer construct comprised of a center bone, muscle, and outer fat layer to mimic thigh tissue properties. This was selected as an initial, simple proof of concept tissue choice. The bulk of the phantom was a gelatin base supplemented with flour, a non-soluble particulate, and evaporated milk, to modulate the echogenic properties of the tissue. This approach allows for increasing or decreasing the flour concentration when other tissue types need to be mimicked. However, this resulted in a uniform construct, quite unlike physiological thigh tissue with muscle and ligament fibrous microstructures creating heterogeneity. Towards that, the high and low flour concentration agarose components added to the tissue phantom were successful at creating a unique non-uniform phantom composition. Obviously, this approach cannot mimic the physiological organization commonly seen in live tissue, but the agarose complexities created a much more challenging ultrasound visualization model. This complexity was highlighted with the addition of shrapnel embedded in the tissue phantom. In a simple phantom without the agarose, flour, and milk components, identifying shrapnel is trivial, but in animal tissue and the phantom developed here, that was not the case.

Using this tissue phantom, different training image sets were configured to evaluate training accuracy in blind swine images. Using a previously developed ShrapML image classification network [31], training with 100% swine images resulted in 99% accuracy, suggesting the model can be trained for identifying shrapnel in animal tissue with its current architecture. However, this was only proof-of-concept, and, for a model fully ready for integration with ultrasound hardware, would require tens of thousands additional swine images to account for sufficient subject variability. To supplement the shrapnel training set, the phantom was designed to mimic the biological complexity, heterogeneity faced in shrapnel detection in physiological tissue, but 100% phantom trained ShrapML models yielded only 61% accuracy in blind swine test image sets.

While the phantom alone cannot replace animal image sets entirely, it exhibited a strong utility in supplementing training sets to reduce the overall animal image requirement. Supplementing the swine images with phantom images so that less total number of animal images would need to be collected, we reached similar accuracies with as little as 25% of the required training images being swine. Further, with as little as 10% of the required images as swine, accuracy was still near 90% in test swine image sets. We compared ShrapML training performance with similar hybrid training ratios with a widely used MobileNetv2 [37] model, as we previously demonstrated it could be trained to achieve high accuracy for shrapnel detection with higher computational efficiency than other models [36]. Training trends were similar for all hybrid ratios except when only 10% of the images were swine, as MobileNetv2 had a much lower accuracy (69%) when compared to ShrapML (87%). ShrapML’s stronger training performance could be due to training challenges when more phantom images were present due to the large difference in number of trainable parameters between the two models (17.2 million ShrapML vs. 3.5 million MobileNetv2). The additional trainable parameters in ShrapML could provide additional opportunities for ShrapML to identify unique features within datasets to correctly correlate classification outcomes. Overall, this improvement, achieved by supplementing with phantom images, could dramatically reduce image acquisition requirements from animals, which are often challenging or costly to acquire. Deep-learning algorithms for more complex applications involving multiple classification categories [38,39,40] will require even more extensive image collection and reducing the required image number by 90% will be substantial.

While supplementing the training set with phantom images reduced the number of swine images needed for reaching strong test accuracies, there are some limitations with the current phantom and training methodology. First, more swine and phantom images are needed to account for enough subject variability to allow this to be implemented with US instrumentation for real-time shrapnel detection. This will also allow for further evaluating the benefit of the tissue phantom by ratio training with larger data sets compared to a reduced number of swine images to fully evaluate the benefit the phantom is providing. For the phantom, the complexity of the phantom can be further improved to better mimic tissue. The heterogenous structure of the phantom made for a variable, challenging training set, but more attention to matching ultrasound properties or integration of key physiological landmarks may further improve on this study. For validating the platform’s echogenic properties, traditional ultrasound properties such as attenuation coefficients, impedance, and speed of sound can be measured using various specialized test setups [41,42,43,44] to better assess how similar the tissue phantom is to physiological soft tissue. In addition, we will look at integration of femoral neurovascular features or translation into other more complex anatomies, such as the thorax and abdomen. A critical task in combat casualty care for ultrasound is the extended Focused Assessment with Sonography for Trauma (eFAST) examination for detecting fluid in the lungs or abdomen. A similar phantom development methodology and subsequent training could be used for tasks such as this, as we have already shown for pneumothorax [45], to create a more complex training environment for training neural network models.

## 5. Conclusions

Deep-learning models have the potential to simplify and reduce the expertise threshold for collecting and interpreting ultrasound images, expanding its use case further into emergency or battlefield medicine. However, as neural network architectures often require large, variable data sets to properly develop models, the approach shown in this work to supplement the training process with easy to obtain tissue phantom images can potentially accelerate model development. In this work, shrapnel detection accuracy in swine images was 96% with 75% fewer animal images or 87% with 90% fewer animal images, due to supplementation with phantom images. Additional work is needed to build more robust tissue phantoms and human or animal data sets for comparisons, but a similar methodology may be translated to other medical imaging applications where adequate image numbers may be hard to obtain.

## Figures and Tables

**Figure 1 jimaging-08-00270-f001:**
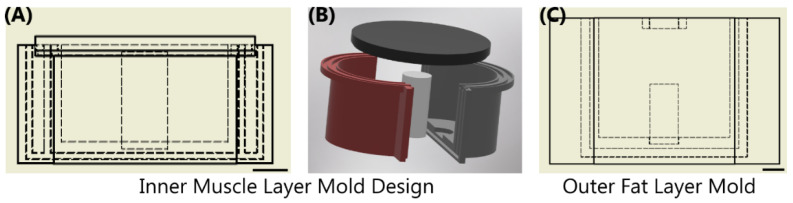
Overview of Sequential Mold Setup. (**A,B**) The inner muscle mold was comprised of 4 components shown as a (**A**) cross-sectional view with inner dimensions shown as dashed lines and (**B**) in exploded view. Two shells were snapped together with an inner bone; top lid was assembled after the phantom solution was added. (**C**) The outer fat layer mold shown with the same bone in the middle of the drawing. Scale bars denote 20 mm.

**Figure 2 jimaging-08-00270-f002:**
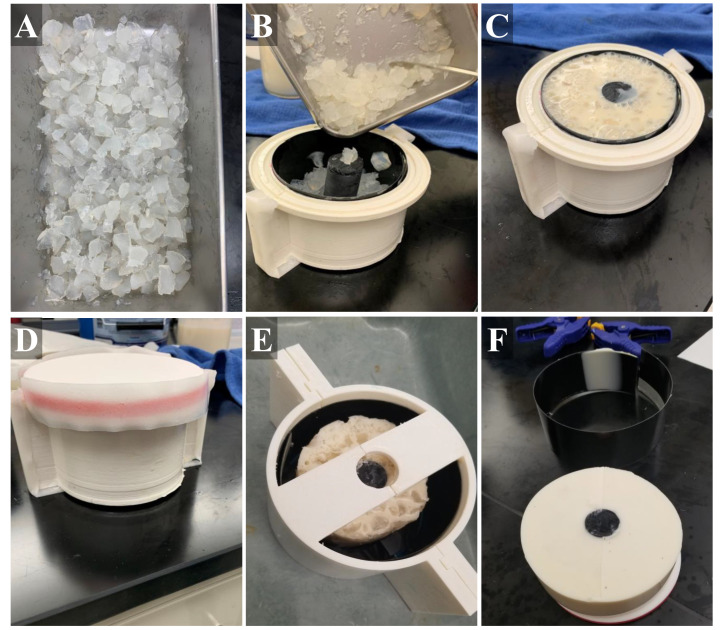
Stepwise construction of the gelatin phantom. (**A**) Chopped agarose fragments with different flour concentrations were first created. (**B**) Next, black plastic liner was placed into the inner mold (muscle layer) and the agarose fragments were added, as well as a plastic bone. (**C**) Then, muscle gelatin solution is poured into the inner mold. (**D**) The lid was secured with tape and the sealed mold placed on ice. (**E**) After solidification, the muscle layer was removed from the inner mold and placed in the center of the outer mold (fat layer). (**F**) After outer layer solidification, the tissue phantom (with bone, muscle layer, fat layer) was removed from the mold and used for imaging applications.

**Figure 3 jimaging-08-00270-f003:**
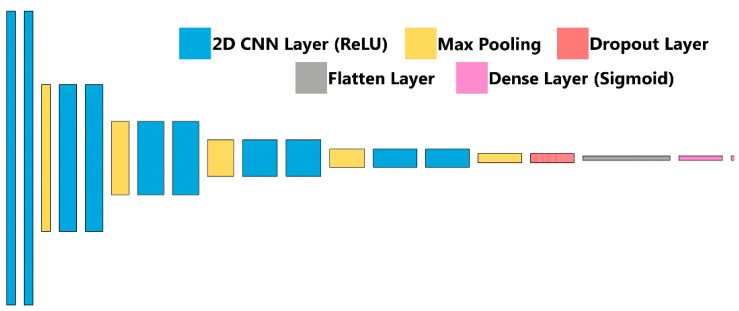
ShrapML Architecture. Block diagram for visualization of the neural network architecture for ShrapML used in training the shrapnel detection models.

**Figure 4 jimaging-08-00270-f004:**
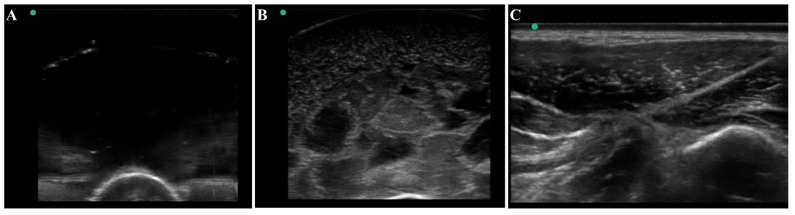
Ultrasound view of the tissue phantom compared to swine tissue. Representative ultrasound images for (**A**) plain gelatin phantom with no additional components, (**B**) the developed tissue phantom with evaporated milk, flour, and agarose fragments, and (**C**) swine tissue.

**Figure 5 jimaging-08-00270-f005:**
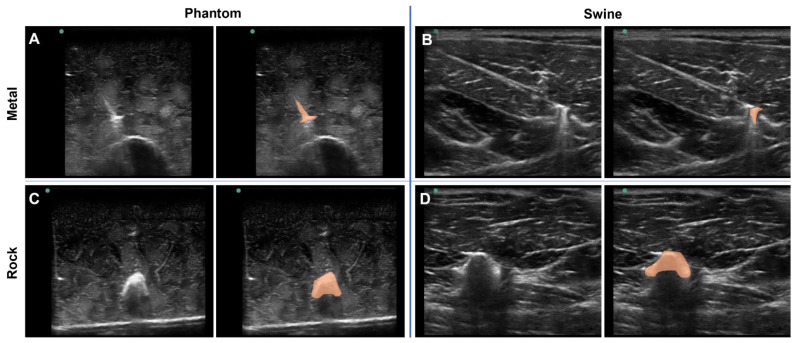
Shrapnel Embedded in Phantom. Representative ultrasound images for metal shrapnel in phantom (**A**) and swine (**B**), and for rock shrapnel in phantom (**C**) and swine (**D**). Right images are replicates of the left image in each panel with segmentation mask overlays (orange) manually drawn to better identify the location and shape of the shrapnel.

**Figure 6 jimaging-08-00270-f006:**
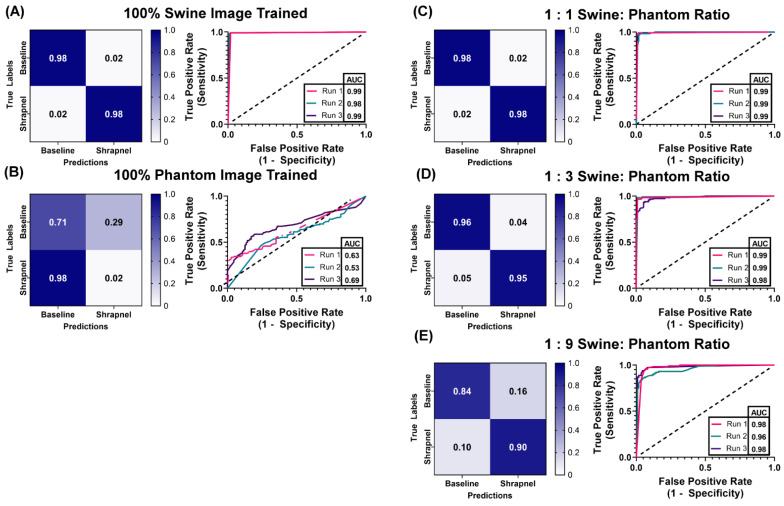
Swine-Phantom Hybrid Training Results with ShrapML for Shrapnel Image Classification. Confusion matrices and ROC analyses for swine image prediction after training with image sets augmented with (**A**) 1:0, (**B**) 0:1, (**C**) 1:1, (**D**) 1:3, or (**E**) 1:9 swine: phantom ratios. The diagonal black dashed line in the ROC plots represents the result for random chance (AUC = 0.50) for reference.

**Figure 7 jimaging-08-00270-f007:**
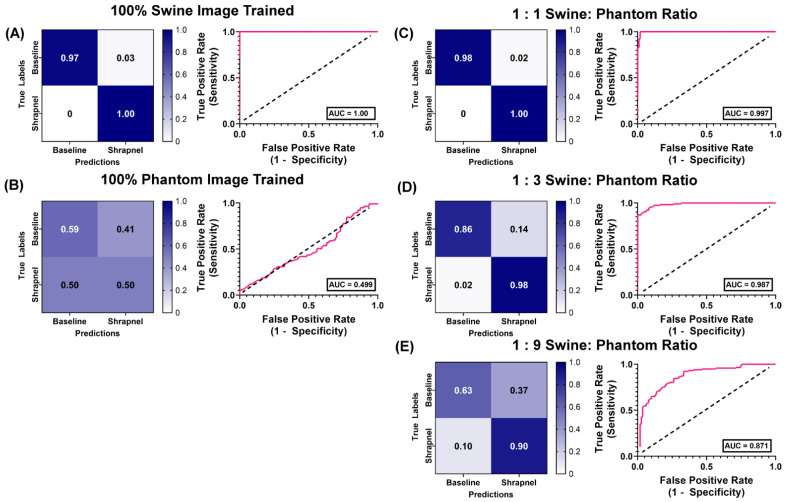
Swine-Phantom Hybrid Training Results with MobileNetv2 for Shrapnel Image Classification. Confusion matrices and ROC analyses for swine image prediction after training with image sets augmented with (**A**) 1:0, (**B**) 0:1, (**C**) 1:1, (**D**) 1:3, or (**E**) 1:9 swine: phantom ratios. The diagonal black dashed line in the ROC plots represents the result for random chance (AUC = 0.50) for reference.

**Table 1 jimaging-08-00270-t001:** Summary of ShrapML Architecture and Training Parameters.

Parameter.	Value
Total # of trainable parameters	17.17 million
Number of Sparsely Connected CNN Layers	5 CNN Layers
Filters in Each CNN Layer	16/32/64/128/256
Number of Fully Connected Layers	1
Filters in Fully Connected Layer	256
Dropout Rate	55%
Training Optimizer	RMSprop
Number of Epochs	100
Learning Rate	0.001
Batch Size	32

**Table 2 jimaging-08-00270-t002:** Total number of swine and phantom images for each training set used in this study. Five swine: phantom ratios are shown: 1:0, 1:1, 1:3, 1:9, 0:1.

	1:0 Image Ratio (0% Phantom)		1:1 Image Ratio (50% Phantom)		1:3 Image Ratio (75% Phantom)		1:9 Image Ratio (90% Phantom)		0:1 Image Ratio (100% Phantom))	
	Baseline	Shrapnel	Baseline	Shrapnel	Baseline	Shrapnel	Baseline	Shrapnel	Baseline	Shrapnel
**Total**	443	467	428	480	422	486	418	490	415	493
**Swine**	443	467	221	234	111	117	45	47	0	0
**Phantom**	0	0	207	246	311	369	373	443	415	493

**Table 3 jimaging-08-00270-t003:** Summary of ShrapML model performance metrics for each training situation. Results are shown as average for three random training: validation splits.

	Swine to Phantom Training Image Ratio for ShrapML Algorithm				
	1:0 (Swine Only)	1:1	1:3	1:9	0:1 (Phantom Only)
**Accuracy**	0.990	0.950	0.960	0.870	0.610
**AUC**	0.990	0.990	0.990	0.970	0.620
**Precision**	0.990	0.930	0.970	0.870	0.690
**Recall**	0.990	0.990	0.950	0.910	0.520
**Specificity**	0.980	0.910	0.970	0.840	0.710
**F1**	0.990	0.960	0.960	0.880	0.580

**Table 4 jimaging-08-00270-t004:** Summary of MobileNetv2 model performance metrics for each training situation. Results are shown for a single training run for each ratio.

	Swine to Phantom Training Image Ratio for MobileNetv2				
	1:0 (Swine Only)	1:1	1:3	1:9	0:1 (Phantom Only)
**Accuracy**	0.982	0.991	0.908	0.693	0.509
**AUC**	1.000	0.998	0.987	0.871	0.499
**Precision**	0.964	0.982	0.829	0.414	0.901
**Recall**	1.000	1.000	0.979	0.902	0.498
**Specificity**	0.967	0.983	0.858	0.633	0.593
**F1**	0.982	0.991	0.898	0.568	0.641

## Data Availability

The datasets generated during and/or analyzed during the current study are available from the corresponding author upon reasonable request.

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
