# Peer review of "Using an Ultrasound Tissue Phantom Model for Hybrid Training of Deep Learning Models for Shrapnel Detection"

_2313-433X, 2022, doi:10.3390/jimaging8100270_

Round 1

Reviewer 1 Report

This work explains the importance of interpreting ultrasound images for fast diagnosis when there is limited trained radiology expertise and highlights the need of using automated image classification achieved by trained deep learning models.  Considering the large amount of data typically required for deep learning algorithms and the difficulty of collecting sufficient data, this work proposes the use of custom-developed tissue phantom for medical imaging in the domain of shrapnel detection. The resulted neural network is a shrapnel detection algorithm applicable to a particular use case: classifying shrapnel in swine tissue. 

Overall, the paper is well-written, with adequate information on the fabrication steps, ultrasonic imaging, thorough detail on both the model training and performance of the classification algorithm of the proposed ShrapML on the hybrid dataset (i.e., different ratios of swine to phantom training datasets). Lastly, the authors compared the performance of ShrapML to that of MobileNetv2, which was previously identified as a suitable model for such detection application. It shows that ShrapML outperforms MobileNetv2 in detection when it comes to hybrid training images.

Here are some minor comments:

-       Is there a reason why the ShrapML model was trained using TensorFlow/Keras and Python, and MobileNetv2 model was trained in MATLAB? Is it primarily due to the MobileNetv2 model was trained in MATLAB for a previous work? 

-       How many images are there before applying the image augmentation? How many manually generated segmentation masks of the shrapnel were needed as the ground truth for training?

Round 2

Reviewer 2 Report

Thank you for your detailed responses to the review. I agree with all changes made.